

# Heart disease severity level identification system on Hyperledger consortium network

Sasikumar R.[1] and Karthikeyan P.[2]

[1] Computer Science and Engineering, K.Ramakrishnan College of Engineering, Tiruchirappalli, Tamilnadu, India
[2] Information Technology, Thiagarajar College of Engineering, Madurai, Tamilnadu, India

Corresponding author
Sasikumar R.,
sasi1986krce@gmail.com

## ABSTRACT

Electronic Health Records (EHRs) play a vital role in the healthcare domain for the patient survival system. They can include detailed information such as medical histories, medications, allergies, immunizations, vital signs, and more. It can help to reduce medical errors, improve patient safety, and increase efficiency in healthcare delivery. EHR approaches are proven to be an efficient and successful way of sharing patients' personal health information. These kinds of highly sensitive information are vulnerable to privacy and security associated threats. As a result, new solutions must develop to meet the privacy and security concerns in health information systems. Blockchain technology has the potential to revolutionize the way electronic health records (EHRs) are stored, accessed, and utilized by healthcare providers. By utilizing a distributed ledger, blockchain technology can help ensure that data is immutable and secure from tampering. In this article, a Hyperledger consortium network has been developed for sharing health records with enhanced privacy and security. The attribute based access control (ABAC) mechanism is used for controlling access to electronic health records. The use of ABAC on the network provides EHRs with an extra layer of security and control, ensuring that only authorized users have access to sensitive data. By using attributes such as user identity, role, and health condition, it is possible to precisely control access to records on blockchain. Besides, a Gaussian naïve Bayes algorithm has been integrated with this consortium network for prediction of cardiovascular disease. The prediction of cardiovascular is difficult due to its correlated risk factors. This system is beneficial for both patients and physicians as it allows physicians to quickly identify high-risk patients and easily provide them with patient severity level using feature weight prediction algorithms. Dynamic emergency access control privileges are used for the emergency team and will be withdrawn once the emergency has been resolved, depending on the severity score. The system is implemented with the following medical datasets: the heart disease dataset, the Pima Indian diabetes dataset, the stroke prediction dataset, and the body fat prediction dataset. The above datasets are obtained from the Kaggle repository. This system evaluates system performance by simulating various operations using the Hyperledger Caliper benchmarking tool. The performance metrics such as latency, transaction rate, resource utilization, *etc.* are measured and compared with the benchmark.

## INTRODUCTION

Different aspects of human survival are significantly influenced by the technological world's explosive growth. It gradually improves comfort and convenience in everyday activities. The advancement in technology helps us in many sectors such as healthcare, social responsibilities, smart appliances, the economy, communication sector, gaming, and many more. The traditional way of treatment has been slowly upgraded with Electronics Health Records (EHRs). The term Electronic Health Record (EHR) refers to an electronic version of a patient's medical history that is kept by a provider throughout time. They may contain all the essential administrative and clinical data necessary for that person's treatment (*Purohit et al., 2021*). It allows multiple healthcare professionals to access, view and edit a patient's medical information within a single, or multiple organizations. EHR sharing is the exchange of patient health information between two or more health care providers. This is done electronically, allowing health care providers to access patient records quickly and efficiently. Electronic health records can also be shared with insurance providers, care coordinators, and other health care organizations to ensure continuity of care. The security and privacy of EHRs are of utmost importance, as they contain sensitive patient information that must be kept confidential. Modern technologies also raise some drawbacks with the above pros. However, disclosing medical records to unknown or unauthorized parties leads to privacy issues and makes a threat to human lives (*Uddin et al., 2021*).

The Internet of Things (IoT) is transforming the health care industry by connecting medical devices and equipment, providing real-time patient monitoring, and enabling predictive analytics. With the help of it, health care providers can now collect and analyze data from a variety of sources and use it to make better decisions, more quickly. IoT-enabled devices can also monitor and track patient vital signs, helping clinicians identify potential problems before they become more serious. Mostly, these records are kept and maintained by centralized third-party authorities and hence, it leads to security threats (*Dash, 2020*). In some situations, the medical records need to be shared among multiple organizations. In these situations, data sharing agreements may be needed to ensure the proper handling and sharing of medical records. The agreement should outline the parties responsible for the protection of the data, the purpose of the data sharing, and the procedures for data access, use, and deletion. The agreement should also include details on how the data will be secured, such as encryption, authentication, and authorization protocols, as well as how any data breaches will be handled. The sharing of confidential information among numerous healthcare suppliers creates security, privacy, and interoperability concerns. To address the above concerns, Blockchain has the potential to expand healthcare by placing the information of the patient at the center of the system and enhancing the health data protection, privacy, and interoperability (*Yaqoob et al., 2022*).

Blockchain technology has been widely accepted by industry applications, due to its characteristics like decentralization, immutability, and distributed ledger (*Nakamoto, 2009*). Many applications have created various blockchain platforms, including Bitcoin, Ethereum, IBM Blockchain, and Hyperledger. Among all the platforms, Hyperledger fabric works with a modular-based permissions network configuration. This permissioned network does not completely trust each other participants for ensuring transparency (*Hyperledger, 2021*). In this network, all entities must have verified identities participate. The participating peers in this network can be any of the following three distinct roles: Endorser, Orderer, and Anchor. The Orderer peer is responsible for ordering transactions and creating blocks of transactions to be distributed to the peers on the network. Another responsibility is to be ensuring that all transactions are properly validated, ordered, and distributed according to the rules of the network. The Orderer peer also ensures that all peers on the network have the same view of the ledger, which is essential for maintaining the integrity of the network. Therefore, the private network guarantees consistent values across multiple organizations (*Ma, Jo & Park, 2020*).

The main goal of this system is to implement attribute-based access control mechanisms to control access to resources by granting or denying access depending on the set of attributes associated with the user, the resource, and the environment. This type of access control can be used to ensure that users only have access to resources and provide granular control over what resources a user is able to access and when (*Nazerian, Motameni & Nematzadeh, 2019*). The availability of the data is guaranteed by this system. As a result, for the patients' vital interest, the access control system cannot deny any legitimate request. In the current scenario, controlling human disease is a challenging task in the medical field. Moreover, disease prediction at an early stage provides a better resolution for human life. Machine learning techniques have been demonstrated for different disease predictions by many researchers. In the present work, a system has been proposed to predict heart disease using Gaussian naïve Bayes (GNB) algorithm. It is a probabilistic classifier that makes predictions based on the probability of an event occurring, given the evidence available. In the case of predicting heart disease, GNB would consider factors such as age, gender, cholesterol levels, blood pressure, *etc.* to calculate the probability that a given patient has heart disease (*Mohan, Thirumalai & Srivastava, 2019*). The algorithm would then make a prediction based on the probability it calculates. Calculating the severity of a heart disease is important to determine the prognosis of a patient and the best treatment options. Knowing the severity of the disease can help a doctor decide what type of medication or procedure is needed and can help patients understand the risks and benefits of each option. It can also help identify the risk factors that need to be managed to reduce the chances of a heart attack or other serious health conditions. This system uses a feature weight computation approach to calculate the heart disease score to prioritize the patient. Weight is an important risk factor for heart disease as it is closely linked to many other risk factors, such as high blood pressure, diabetes, and high cholesterol. Weight calculation can also help healthcare professionals determine the severity of heart disease in an individual and assess the effectiveness of treatments.

The proposed system's main contributions are as follows:

- Dynamic attribute-based access control policy in the Hyperledger consortium network has been implemented using a combination of chaincode functions. The chaincode functions help in the creation of access permissions for different people and organizations, while smart contracts are implemented to enforce the access control policy. This approach helps to control access to resources such as patient medical information.

- The naive Bayes algorithm is used to predict the presence of a heart disease by evaluating the probability of the presence of a heart disease from the given datasets. The algorithm, which is implemented in a chaincode function, will compute the probability of the presence of a heart disease and the chance of the absence of a heart disease and make a prediction depending on which probability is higher.

- Finally, the system implements the feature weight computation approach in the fabric chaincode function. This approach involves computing the weights of different features that can be used to determine the severity of the disease. The higher the weight of a certain feature, the more severe the heart disease is.

The rest of this article is structured as follows: The literature review section presents current research on blockchain based EHR solutions. The proposed system section describes the structure of the system, components, module implementations, and smart contract functions. The result section describes the simulation setup, system performance evaluation using various metrics, and comparison with existing works. The final section concludes and discusses future work.

## LITERATURE REVIEW

This section discusses the most recent developments in healthcare systems that use blockchain technology, especially those developed for the secure exchange of medical data.

*Tagde et al. (2021)* conducted a thorough examination of the blockchain concept in an e-Health system using Artificial Intelligence (AI). With a large volume of data, AI employs a variety of algorithms and decision-making capabilities. They highlighted how AI can help blockchain technology increase service efficiency in their study. They came to an agreement on how AI and Blockchain integration would minimize cost and improve efficiency (*Tagde et al., 2021*). *Meier et al. (2021)* have implemented a blockchain system to manage and transact Personal Health Records with data security and data privacy concerns. This system makes it easier for patients and medical practitioners to share health information on a demand basis. Also, this system ensures that the stored personal information can be accessed only by an authorized member using a key management server (*Meier et al., 2021*). *Vora et al. (2019)* have expressed that blockchain is one of the best technologies to store and manage Electronic Health Records in an efficient manner. Further, the authors have described, that the patient's confidential information can be shared among different parties in the medical field without compromising security and privacy. *Sammeta & Parthiban (2021)* have proposed a Hyperledger architecture to control

patient EHR data with participants of a network. This model allows users to control their data and give authentication to third parties to read/write. This system uses the SIMON method for the encryption of user information and a Group Teaching Optimization Algorithm (GTOA) has been applied to improve the performance. Besides, this system addresses one of the blockchain limitations such as handling a large volume of data (*Sammeta & Parthiban, 2021*). *Wang et al. (2019)* have framed a consortium blockchain architecture for the EHR sharing protocol. The data requester can make requests on EHR data, and the requested data will be given to the requester in the form of cipher text from cloud storage. In the future, the requester can re-encrypt the provided data with prior approval of the corresponding data owner. To guarantee the availability of the system, the proposed system uses proof of authorization as a consensus algorithm (*Wang et al., 2019*). *Wang & Qin (2021)* have studied interoperability and privacy framework in health care data sharing. In addition to the fabric build-in privacy approach, the proposed architecture implements smart contract-based hierarchical access control to provide enhanced privacy. This system has been adopted with permissioned blockchain and smart contract to overcome the existing privacy issues in healthcare data. Each node or member of this network needs to get prior approval for executing the transaction (*Wang & Qin, 2021*). Classical data sharing systems face security issues since they rely on centralized storage. Currently, numerous researchers are implementing a blockchain-based electronic health record system which delivers security and traceability. To give individuals who own data the ability to customize the authentication process, this system incorporates attribute-based encryption. To reduce the expense of data storage, it uses both off-chain and on-chain storage strategies (*Ray et al., 2021*).

*Nazerian, Motameni & Nematzadeh (2019)* proposed Emergency RBAC (E-RBAC), this allows the system to differentiate between different levels of access rights for different users. It allows emergency responders to have access to patient data when needed, while also maintaining security and privacy. *Tamilarasi & Jawahar (2022)* have proposed a lightweight encryption algorithm to secure data storage and access management. To reduce the computational complexity in terms of execution time and energy consumption, the proposed system integrates Paillier encryption with the KATAN algorithm. Usually, lightweight encryption techniques are unable to provide better security due to their key-space size. To resolve this, the study proposes a swarm optimization algorithm. The Swarm Optimization Algorithm (SWO) helps to randomize the number of iterations in the key management process. The framework was demonstrated as secure and reduced error rates (*Tamilarasi & Jawahar, 2022*). *Lavanya & Kavitha (2022)* have proposed a cloud assistant blockchain for securing Electronic Health Records. This system uses the SecPAKE protocol to generate key based access control to secure EHR data. To ensure the enhanced security of the cloud-based blockchain, the proposed system adopts biometric traits with the key. The SecPAKE protocol includes EHR record access policies. To prevent server impersonation attacks, every transaction verifies the user authorization with biometrics. A detailed evaluation of security flaws in the medSupport blockchain prevents any attack from taking place. Additionally, it performs better in terms of computational cost, latency, and other performance parameters (*Lavanya & Kavitha, 2022*). *Chelladurai & Pandian*

*(2022)* developed a smart health paradigm that would allow patients' health information to be shared among healthcare providers. Smart contracts enable patients to register, update, share, and access the information. The transactions and blocks were hashed with SHA-256 in the specified model workflow. By using hash functions, this system achieves transparency, security, and integrity. Immutable logs are used to ensure integrity in this case (*Chelladurai & Pandian, 2022*). *Huang, Ma & Zhang (2019)* presented a study on the consensus protocol, which would be a major component of blockchain technology. This protocol embodies the performance and functionality of the blockchain system. The characteristics such as packet loss and election timeout have been evaluated with respect to network size. Finally, they concluded that Raft would be a well adopted consensus algorithm for private blockchains (*Huang, Ma & Zhang, 2019*). *Wang & Jin (2011)* proposed the discretionary access control (DAC) as user-centered access control methods. The data creator defines access control policy and implemented using an access control matrix. Only simple environments are acceptable for this strategy (*Wang & Jin, 2011*). The author has described the proposed healthcare system, which integrates blockchain and the Internet of Things. To acquire health information, this system makes use of multiple IoT sensor nodes, including temperature, galvanic, heart rate, oxygen saturation, and a few more. To ensure privacy and security, the system employs a hybridized security system with a dual layer to monitor and manage the flow of EHR data between data provider and requester. Finally, the authors were able to show that the proposed model performed effectively in several kinds of blockchain-IoT situations in terms of availability, flexibility, and security (*Wang & Guan, 2023*).

For preserving privacy and security, most of the work involves blockchain-based EHR systems. To make the healthcare system smarter, it is crucial to integrate prediction algorithms within the blockchain framework. To build a smart healthcare system, this research incorporates ABAC mechanism to protect sensitive data in electronic health records, heart disease prediction algorithm using chaincode and delivering care to patients in accordance with their severity level.

## PROPOSED SYSTEM

This section explains the working principles of the proposed Hyperledger blockchain system for the exchange of healthcare information. For global patient treatment, the system requires efficient EHR storage and retrieval. This architecture includes two functional phases: sharing EHR data among several organizations and calculating the severity level of heart disease. The EHR data sharing system model is illustrated in Fig. 1.

This paragraph provides a brief explanation of how user registration and EHR data were exchanged across several organizations. In the user registration phase, the administrator of an organization must enroll and obtain their corresponding identity from the Fabric CA to participate in the fabric network. After that, the healthcare participants must register and obtain their member identity (MID) from the CA through the organizational administrator. To store resources securely, the data owner can now register with an organization using asymmetric key encryption. In EHR data sharing phase, the data requester can submit a request for resources by providing their user attributes as well as

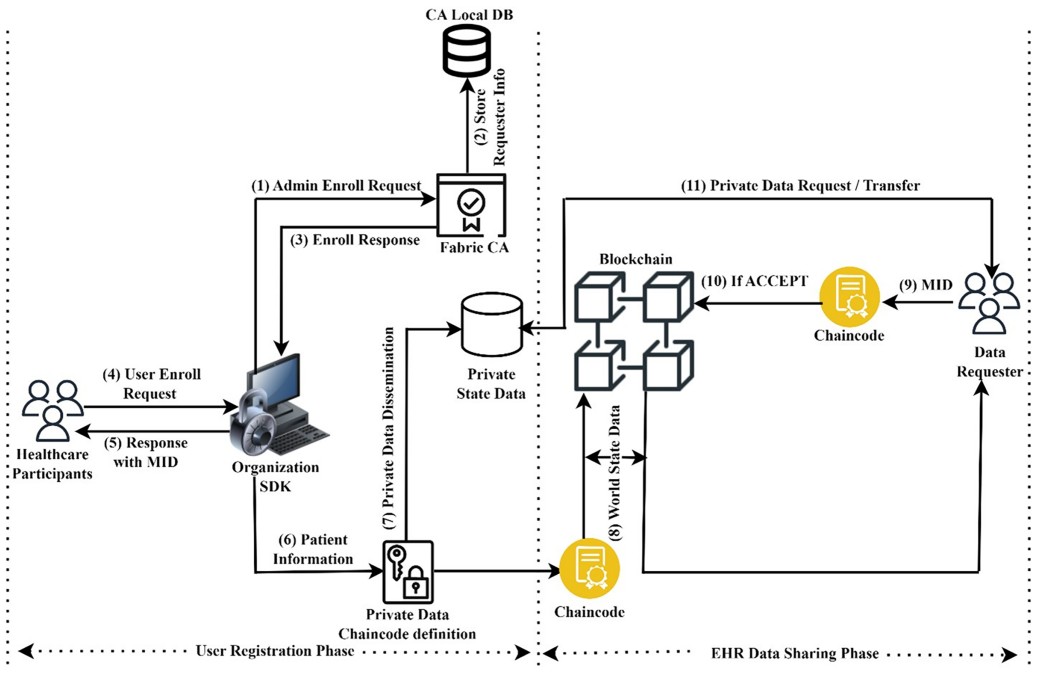

**Figure 1 Sharing of secure EHR data.**           

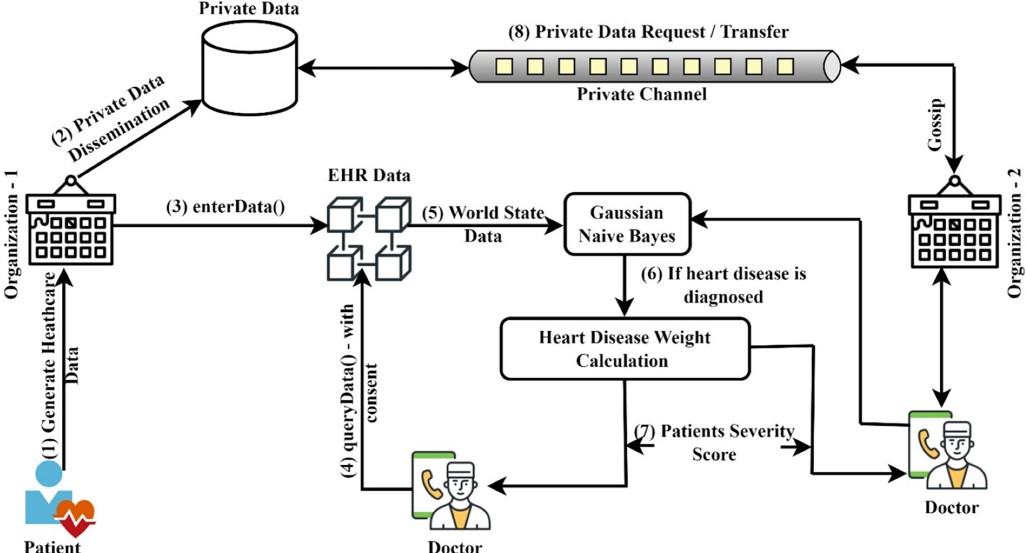

**Figure 2 Model for predicting heart disease and determining its severity.**

their identities. For example, a doctor may be able to view certain medical tests, but not make changes to the results. User identity and enrolment process section provides a complete description of how to register as members on the fabric network and transfer EHR data securely.

**Table 1 The notations in this article.**

| Notation | Description |
|---|---|
| CA | Certificate authority |
| RK | Randomly generated symmetric key |
| DB | Database |
| PK | Private key of an organization |
| PID | Patient Unique Identity |
| Encr | Encryption of data using a specific key |
| MID | The term MID refers to a member's unique identity |
| Dsign | Digital signature of data, which is signed by admin |
| secKey | A member's secret key |
| r | Refers to resources such as a patient's medical records/Resources |
| $\parallel$ | Evaluates whether either policy has been satisfied |
| R1, R2. . . Rn | Roles and permitted operations for users 1, 2…n |
| $Policy_{sub}$ | The data requester's authorization to access data resources |
| $\alpha$ | A user's unique attributes |
| $\widehat{\ }$ | Requester must conform to the organization's policies, data policies, and allowed actions |
| T1 | Starting time for data requests |
| T2 | Ending time for data requests |
| $W(a_n)$ | Weight of a feature in relation to attribute $(a_n)$ |
| $R(a_n)$ | Range of a selected attribute $(a_n)$ |
| $D(p)$ | Severity level of patient 'p' |
| ET | In case of an emergency, a team was created to get patient records |
| PR | Refers to the access control policy rule |

Figure 2 illustrates the general procedure of heart disease prediction and severity level identification. The naive Bayes classifier was implemented in this system to predict heart disease from the datasets submitted. This system focuses on determining which patients are more prone to have heart disease based on a variety of medical factors. The ledger query will be used to communicate the stored patient data to the Gaussian naive Bayes smart contract function. The purpose of this smart contract is specifically to predict heart disease. The patient data is provided in the GNB smart contract, which will consider the attributes to forecast the risk that patients may have heart disease. The work has been divided into four modules. The first module's objective is to use the smart contract function to authenticate the registration of healthcare participants in an organization. The second module includes the patients' basic functionalities, including registering, updating, and viewing their personal and medical history. Authentication and privacy for network users are provided by these two modules using access control policy and private data techniques. In the third and fourth modules, it is discussed how to predict heart disease using the chaincode function and how to calculate severity score using the supplied datasets. The next sections discuss the various components of the proposed system, the

access control policies that have been applied, and the chaincode functionalities. The symbols and operations used in this article are shown in Table 1.

## Component's roles and responsibilities of the proposed system

The proposed system network has been designed using multiple organizations as a consortium blockchain. Through peer-to-peer connections, the system enables collaboration between various organizations such as hospitals, clinical laboratories. The Raft consensus protocol is used to create this system, and it operates in a leader and follower pattern. The Raft consensus algorithm has been implemented to provide Crash Fault Tolerant (CFT) and enhanced understandability in a distributed network. This system has a special node called the Orderer node which is the central part of the fabric network. The Orderer node's responsibility is to maintain a consistent database, distribute and ensure consistency among multiple peers in the participating organization. Three ordering nodes make up this system, where a leader node will be dynamically selected from among them. The reason for having more than one Orderer node is to achieve fault tolerance in this network. The role of the membership service provider (MSP) is to supervise and validate participants' identities, provide crypto materials for network entities during transaction proposals and to revoke user identities with the help of Certificate Authorities (CA). The CA facilitates user enrollment and transaction invocation in blockchain networks by providing necessary certificates. The next key component is the channels or ledgers, and it is a subnetwork of a network that helps to participate in the form of proposing and receiving transactions from their own peers. The transaction details of any channel can be viewed only by members of a corresponding channel (*Hang & Kim, 2021*; *Manevich, Barger & Tock, 2018*).

A world state database, smart contracts, and a copy of the ledger are all present to each peer. The main role of the peer is controlling the ledger in the form of reading/writing by chaincode execution. The categories of peers and their roles are: (1) Endorsing peers—using its copy of the ledger, it executes the proper smart contract in response to the client's request. (2) Committing peers—for future audit purposes, these peers analyze each transaction request and update its copy with either valid or invalid transactions. (3) Anchor peers—across the peers in the organization, anchor peers disseminate the block details from the ordering service. (4) Leader peers—these are like anchor peers in that they also disseminate block information, however this leader peer does it using the Gossip protocol.

## Private data and endorsement policy

To ensure the organization-level privacy of patients' medical records, in the proposed system, the private data collection concept has been utilized. These private data are accessed in a peer-to-peer manner between the authorized organizations using gossip protocol (*Benhamouda, Halevi & Halevi, 2018*). The patient's medical records can be endorsed by an organization specified in the private collection. Other healthcare centers are not able to access the patients' health information stored in a different hospital even though they are in the same communication channel. The definitions for the hospital's

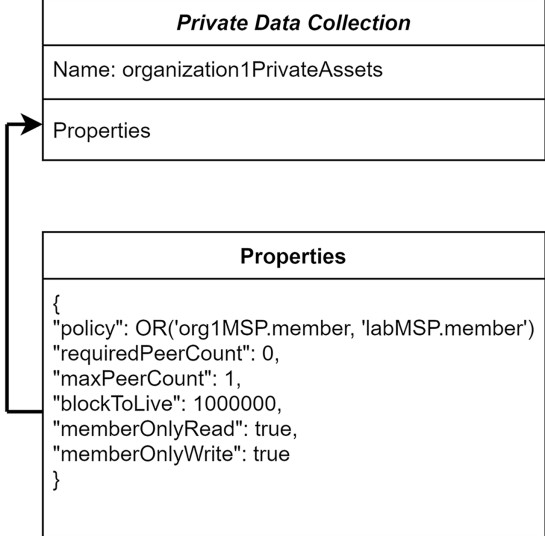

**Figure 3** **Organization's private data collection.**

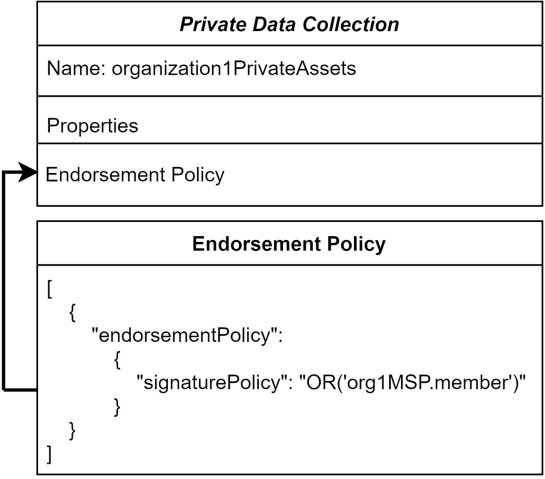

**Figure 4** **Collection level endorsement policy.**

private data collection are shown in Fig. 3. The consideration of policy in "Hospital1PrivateCollectionAssets" allows the members of Hospital-1 and Laboratory-1 to access and transact the data.

The ordering service transactions do not comprise private data. So, these private data are not distributed among all the peers in this network. To reflect the details of private patient data to the authorized organization and its peers, endorsing peers takes part in a major role. Figure 4 illustrates the endorsement policy for the private collection. It defines any transaction executed in the private data should be endorsed by Hospital-1 through endorsing peers.

## User identity and enrolment process

This helps to ensure that only authorized people have access to sensitive patient information. The proposed system is specifically designed to provide secure and private access to patient records. It uses digital identity and attribute management to ensure that only authorized individuals can access sensitive information. These identities are signed by a trusted entity, usually a CA, and used to represent entities in the network. It enables organizations to assign specific permissions to entities on the network, and to define the conditions under which these permissions can be granted or revoked. It allows each organization to create roles and assign privileges according to healthcare professional needs. Each network organization has a registered administrator with responsibility of managing users and confirming the identification and attributes with CA. On the fabric network, the user identity registration process is split into two phases: the organization admin registration and the healthcare user (*i.e.*, doctors, nurses, and lab technicians) registration.

### Administrator identity registration

Transparency in transactions is the primary goal of Hyperledger. As a result, installing and/or instantiating queries at this time is only permitted for the Admin. The following steps need to do Admin identity registration:

1. An administrator of an organization submits an enrollment request to Fabric CA, which contains an encrypted format of admin identity and random key 'RK'.

$$enrollReq \leftarrow \left\{ Encr_{CA-Public_{Key}}(AdminInfo, RK) \right\}$$

2. For further verification of the previously stated request, the CA maintains the administrator information in its local database, *i.e.*, $CA_{DB}$ and issues the identity certificate for the admin with the private key. The admin will then use a private key to decrypt the response after first applying the CA's public key to verify the signature.

$$CA_{DB} \leftarrow Verify\left\{ Decr_{CA-Private_{Key}}(enrollReq) \right\}$$
$$enrollResponse(Org_{Admin}) \leftarrow Encr(Admin_{Cert}, Admin_{PK})$$

where, PK refers to the private key of an organization Administrator.

### Healthcare user identity registration

The following steps need to do user identity registration on the fabric registration:

1. Healthcare users who wish to join the network must submit a request to the organization administrator along with user information, such as identity, attributes, and randomly generated symmetric key 'RK'.

$$AdminReq \leftarrow memberRegisterReq\left\{ memberInfo, Encr(CA_{Public_{Key}}(RK)) \right\}$$

2. It assigns member identity (MID) when member details have been validated and approved, after which it provides CA, the encrypted member data along with the assigned MID.

$$CA_{Req} \leftarrow memberRegisterReq\left\{Encr_{(CA-Public_{Key}}(memberInfo, MID)), Dsign_{Admin}(memberRequestInfo)\right\}$$

where, Dsign denotes the digital signature signed by admin using their private key.

3. Consequently, CA verifies its signature using admin's public key. When a request is passed during the verification process, CA decrypts it uses its own private key to store the member's identification, personal information, and a random key.

$$CA_{DB} \leftarrow verify_{Admin_{public-key}}(CA_{Req})$$

The secret value that the CA issues to the requested user is made up of the user's secret key, the administrator's identity certificate for the required organizations, *i.e.*, hospital, and associated member role.

$$memberResponse \leftarrow \{Encr_{RK}(Identity_{Admin}(Org), role, secKey), Dsign_{CA}(registerResponse)\}$$

where, secKey denotes the member's secret key provided by CA, role denotes the corresponding client role in the proposed fabric network. After receiving the CA's response, the requester uses the public key of the CA to confirm the signature. After submitting an enrollment request to the CA with the secKey, the CA decrypts the request through its private key to acquire the identity of the user. Consequently, this system uses two different certificates for member registration and enrollment process. The first certificate to validate the identities of the users and the second one to authenticate the ownership of an attribute.

### Establishing secure communication between the local database and Hyperledger system

The local database, CouchDB, is set up to use HTTPS/TLS by generating an SSL certificate and a private key. Typically, CouchDB listens on port 443 for incoming requests. In the other part of blockchain architecture, network components such as peers and orderers use certificates issued by common certificate authorities. Finally, TLS mutual authentication ensures better security between local database and blockchain system.

### Attribute based access control policy

Attribute-based access control (ABAC) is a powerful security mechanism for protecting the privacy of EHRs when sharing them between different healthcare organizations. It allows for the secure exchange of EHRs between different organizations by using attributes, such as an individual's identity, role, and access rights, to determine who can access and view the EHR. The policies can also be set to dynamically adjust based on the attributes associated with the participating user and medical records. For example, an ABAC system may grant access to an EHR if the user is a doctor in the emergency department of a hospital but deny access if the user is a doctor in the primary care department. ABAC provides the flexibility to define policies that are tailored to the particular use case and can be modified as the security requirements change (*Shammar, Zahary & Al-Shargabi, 2022*). The core elements of attribute-based access control in EHRs are as follows:

- **Attributes (or) subjects:** These are the characteristics used to define user access. Subjects can include things like organizations participating member's role (like doctors, nurse), department, and other user information.
- **Objects:** It represents various resources, like patient's demographic information, medical history, medications, vital signs, laboratory tests, clinical notes, and insurance information.
- **Operations:** It includes assigning roles to each user, determining user access levels (like view, edit, or delete), set up notification systems and assign privileges to each user (such as creating a new patient record or scheduling an appointment).
- **Policies:** Policies are used to define which users can access which resources. Policies can be based on user attributes, or they can be based on context, such as the time of day or the type of device being used.
- **Environments:** All access to EHRs must be logged, and the log should be reviewed on a regular basis and Patients should be able to specify which locations, if any, EHRs can be accessed from.

When considering shared EHR data, attribute-based access control systems offer the following advantages:

- **Fine grained authorization:** It uses user attributes, such as user roles and group membership, to define access rights to EHR data. This type of access control allows for very specific authorization decisions to be made, based on the user's specific attributes. For example, a doctor may have access to a patient's medical record, while a nurse may only be able to view the patient's list of medications.
- **Independent authorization:** Attributes can include the patient's identity, the owners of the data and the receiver of the data, the type of data being shared, and the purpose of the data sharing. With ABAC, each data sharing transaction can be independently authorized, ensuring that only the necessary and appropriate data is shared.
- **Attribute management:** ABAC enables healthcare organizations to manage access to medical records based on the attributes of the user, such as their role in the healthcare organization, their location, or even the type of data they are trying to access. This makes it possible to create complex and nuanced access policies that are tailored to the needs of the healthcare organization and its users.

### Policy creation and management

This system can enable organizations to manage and enforce EHR access policies, such as patient consent and authorization requirements, in a secure and immutable manner. By leveraging the features of Hyperledger Fabric, such as distributed ledger technology, smart contracts, and digital signatures, organizations can create and manage a secure and reliable EHR access policy management system.

- **Policy creation:** Users can create policies in the form of access control lists (ACLs) to specify who can access certain resources. ACLs can be based on user attributes, such as

**Table 2 Healthcare participants' roles and actions.**

| Roles (R) | Permitted actions |
|---|---|
| Doctors | Read, Update |
| Nurse | Read |
| Lab technicians | Update |
| Patients | Read |

their role, or the attributes of the resource they are trying to access, such as the type of data.

- **Policy enforcement:** Access control policies can be enforced using the fabric's chaincode, or smart contract, which is the program that is executed on the blockchain. The chaincode can check the attributes of the user and the data being accessed to ensure that the user is allowed to access the data.
- **Policy storage:** Policies can be stored in the blockchain, allowing them to be shared across different organizations and users, and ensuring that they cannot be tampered with.

A data owner should create an attribute-based access policy in the EHR that is specific to the subject and the environment of the data. The policy should also specify which attributes can be used to restrict access to the data, such as age, gender, medical history, *etc*. The policy should also define how these attributes can be used to determine who has access to the data. By applying the attribute-based access control policy, the new policy (p) can be defined about EHR resources, such as patient medical information (r). In ABAC, multiple policies can be established in relation to both the environmental circumstances and the user (u) who will access the resource (r) from the specified organizations. For example, the following policy can be defined: 'u' can access 'r' if 'u' has the necessary permissions, and the environmental conditions are within acceptable parameters. The various healthcare providers included in the EHR system are described in Table 2, along with their permitted actions.

The policy is a collection of the actions that network members have requested. Depending on the user attributes requested, the operations can either read or write against the given resource.

$$(1) \ Org_1Policy = Policy_{sub1}||Policy_{sub2}||Policy_{sub3}||\ldots Policy_{subn}$$
$$(2) \ Action_{Read} \leftarrow Subject_{(R1||R2||R3\ldots Rn)}(Resource)$$

where, $Org_1Policy$ denotes, set of rules and regulations that govern the activities and operations of the organization. The $Policy_{sub}$ requires that the data requester must have a valid authorization from the data resource owner and must have a valid purpose for accessing the resource. It is important for organizations to have a policy in place to make sure that employees are aware of their roles and what is expected from individuals. The policies should be updated and revised as needed, to ensure that they remain relevant and effective.

---

**Algorithm 1** Create a new policy.

Input: object, policyId, subject, environment

Output: Boolean Status

async definePolicy(object, policyId, subject, environment)

{

> *if* (!*object*||!*policyId*||!*subject*||!*environment*){
>
> > *return false;*
>
> }
>
> *letpolicy* = {
>
> *Id*: *policyId*,
>
> *subject*: *subject*,
>
> *env*: *environment*
>
> };
>
> *if* (!*object.policies*){
>
> *object.policies* = [];
>
> }
>
> *object.policies.push*(*policy*);
>
> *return true;*

}

---

RBAC in EHR also offers environmental controls like time-based access privileges, which is an important determinant. The meaning that a user can be given access to a certain feature for a certain period and then the access will be revoked once the specified period has ended. This helps to ensure that users do not have continuous access to sensitive information that they are not authorized to view. The data owner can manage policy contracts dynamically to ensure that data access rights are controlled in accordance with the most recent requirements. Each user has unique attributes ($\alpha$), such as working organizations, department, and what responsibilities they play. With the help of those attributes, sub policies can create like following:

(3) $Sub_{policy} = \alpha_1 \ AND \ \alpha_2 \ AND \ \alpha_3 \ AND \ ...\alpha_n$

(4) $Decesion_{grant} \leftarrow Org_1 Policy \bigwedge Sub_{policy} \bigwedge Action_{Permitted}$

where, the policy can be expressed as a statement that evaluates to true or false, depending on the values of the attributes. This policy determines whether a user should be granted or denied based on the values of the attributes associated with the action.

### Chaincode construction

The system's blockchain smart contracts are described in this section. These smart contracts can be used to manage the access rights of users to view and modify patient medical records. Algorithm 1 illustrates the process of defining a new policy. It takes input as object, policyId, subject and environment.

---

**Algorithm 2 View the existing policy.**

Input: policyId

Output: Policy

async viewPolicy(ctx, policyId)

{

    *let policyAsBytes = await ctx.stub.getState(policyId);*

    *if (!policyAsBytes||policyAsBytes.length == 0){*

        *throw new Error('{policyId}does not exist');*

    *}*

    *return policyAsBytes.toString();*

}

**Algorithm 3 Update the present policy.**

Input: patientID, policyID, policy

Output: Boolean Status

async updatePolicy(ctx, patientID, policyID, policy)

{

    *let policyAsBytes = await ctx.stub.getState(patientID);*

    *if (!policyAsBytes||policyAsBytes.length == 0){*

        *throw new Error('{patientID}does not exist');*

    *}*

    *let ehrRecord = JSON.parse(policyAsBytes.toString());*

    *if (!ehrRecord.policies[policyID])*

    *{*

        *throw new Error('{policyId}doesnotexist');*

    *}*

    *let updatedPolicy = JSON.parse(policy);*

    *ehrRecord.policies[policyID] = updatedPolicy;*

    *await ctx.stub.putState(ownerID, Buffer.from(JSON.stringify(ehrRecord)));*

    *return updatedPolicy;*

}

Where, policyId represents the identifier of a policy, subject represents the network participating members, environment represents time and emergency status of the patients. Algorithm 2 accepts a policyId as input and uses it to determine whether the policy is available in the chaincode. Algorithm 3 is used to obtain the current policy state using GetState() API and to update the policy rule based on the current scenario. There are two

**Algorithm 4 Delete the policy.**

Input: policyID

Output: Boolean Status

async deletePolicy(policyId)

{

```
        try {
        const policyExists = await viewPolicy(ctx, policyId);
        if (policyExists)
        {
        const deletePolicyRequest = {
        fcn: 'deletePolicy',
        args: [ctx, policyId]
        };
        await contract.submitTransaction(deletePolicyRequest);
        return true;
        }
        else{
                throw new Error('policy does not exist');
        }
        }
        catch (err) {
        throw err;
        }
}
```

scenarios where Algorithm 4 can be used. It can call when the defined policy time has expired and then explicitly call to remove the current policy.

## Smart contract for heart disease prediction and weight computation

This section describes the datasets information, feature selection, the methodology used to diagnose heart disease, and feature weight calculation for ranking the patient. In this phase, the entire work is broken down into four stages. At stage 1, attributes from EHR records are chosen for prediction of heart disease. Stage 2 uses the Gaussian naive Bayes technique to predict the disease. The severity level of cardiovascular disease was determined in stage 3 using a feature weight computation approach. Finally, the Emergency Attribute Based Access Control (E-ABAC) model has been applied for emergency care.

### Datasets and feature selection

In this section, four medical datasets, the heart disease dataset (*David Lapp, 2018*), the Pima Indian diabetes dataset (*National Institute of Diabetes and Digestive and Kidney*

---

**Algorithm 5** Heart disease prediction using Gaussian Naïve Bayes.

Input: EHR Data

Output: Boolean Status

```
async predictHeartDisease(patientData){

    const predictorVariables[] = {patientData.age, patientData.sex, patientData.cp, patientData.trestbps, patientData.chol,
                    patientData.fbs, patientData.restecg, patientData.thalach, patientData.exang, patientData.oldpeak,
                    patientData.slope, patientData.ca, patientData.thal};

    const X_Train, Y_Train = train_test_split(data, analyzevalue, proximatevalue)
    let Y_train = X_train.map((item) => item["target"]);
    let Y_test = X_test.map((item) => item["target"]);
    var mod_X_Train = [], mod_X_Test = [];
        C_XTrain.forEach((key) => {xTra.push(X_train[i][key]); });
        mod_X_Train.push(xTra);
    model = newGuassianNB()
    model.train(X_Train, Y_train);
    let prediction = model.predict(mod_X_Test);
    result = prediction;
    return result;

}
```

Diseases, 2019), the stroke prediction dataset (*Fedesoriano, 2021b*), and the body fat prediction dataset (*Fedesoriano, 2021a*) are used to implement the proposed system. These datasets were obtained from the Kaggle UCI Repository. The first heart disease dataset includes 1,025 instances with 13 attributes. The second diabetes dataset contains 768 instances with eight attributes. Third, the stroke prediction dataset includes 5,110 instances with 11 attributes. Finally, the body fat prediction dataset contains 252 instances with 15 attributes. The four datasets were used to create electronic health record datasets. For the prediction of heart disease, 13 features were considered. Those features are age, sex, chest pain (cp), resting blood pressure (trestbps), cholesterol (chol), fasting blood sugar (fps), resting electrographic results (Rrestecg), maximum heart rate achieved (thalach), exercise induce angina (exang), ST depression induced by exercise (oldpeak), slope of the peak exercise ST segment (slope), number of major vessels colored by fluoroscopy (ca), and the thallium heart scan (thal).

### Heart disease prediction using Gaussian naïve Bayes smart contract

Blockchain and Gaussian naïve Bayes Classifier have been coupled to offer the healthcare sector a better solution. This system focuses on determining which patients are more prone to have heart disease based on a variety of medical factors (*Li et al., 2020*; *Palaniappan et al., 2022*). The task of determining the cause of the cardiac disease is not an easy task. It

**Table 3 Significant features weights.**

| Significant features | Features most occurrences count | Associated weights |
|---|---|---|
| sex | 20 | 0.17 |
| cp | 18 | 0.15 |
| fbs | 12 | 0.09 |
| exang | 12 | 0.09 |
| oldpeak | 14 | 0.12 |
| slope | 12 | 0.09 |
| ca | 19 | 0.18 |
| thal | 14 | 0.11 |

depends on many risk elements, and it needs an efficient system to detect in the early stage. Algorithm 5 illustrates the implementation methods for disease prediction.

### Weight calculation using feature weight

Various features have been evaluated to predict the severity of heart disease, and different weights have been assigned to each feature based on its significance. Assigning an incorrect weight to an attribute leads to failure in the treatments. Let us consider 'S' as the set of heart disease attributes, *i.e.*, S = {a1, a2, a3,…., a13 } and each attribute affiliates with different weights {attribute1=value, attribute2=value,..} where 0 <= value <= 1. The basic principle of feature computation is that each feature has different impact factors in disease prediction. The range of attribute impact factors should be 0 to 1. If an impact factor is closer to 1 for a feature, it indicates that this feature is considered as the more important one than others. On the other side, the weight of the features has near 0, which indicates that it is less significant. A total of 13 features has been chosen for the prediction of heart disease, and of them, eight features have been chosen for the severity calculation (such as 'sex', 'cp', 'fbs', 'exang', 'oldpeak', 'slope', 'ca', 'thal'). This feature selection is done by referring to *Yazdani et al. (2021)*'s experimental results. The authors have used different classification models for the same dataset to predict significant features. To find significant features, they carried out a series of experiments with 8,100 distinct combinations of features using seven different classification models. To calculate the severity of patients, only the top performed features of the dataset have been considered. The top performance is identified by referring to the highest accuracy, highest F-measure, and highest precision values of the obtained results, the above eight features have been identified for calculating the heart disease severity.

Table 3 represents selected features, the occurrence counts of the features, and the Weights associated with each feature. The individual feature associated weight is calculated through the following equations:

$$W(a_n) = \frac{o}{\sum_1^n s_0 + s_1 + \cdots + s_n}$$

where, 'W' represents for the weight assigned to the chosen attribute 'a', 'o' stands for the frequency with which the chosen attributes appeared in the experiments conducted by

---

**Algorithm 6  Calculating disease severity score.**

Input: patientID

Output: score

*async weightComputation(patientID){*

*heart_points = 0;*

*selcted_features[] = age, cp, fbs, exang, oldpeak, slope, ca, thal;*

*selected_features_weight = AssociatedWeights;*

*foreach target in input*

*calculate individual_feature_weights;*

*calculate total_weight_of_features;*

*heart_points = total_weight_of_features;*

*list.append(heart_points);*

*end foreach;*

*return list;*

*}*

---

*Yazdani et al. (2021)*, and 'S' stands for the significant attributes considered when predicting the risk of heart disease. If we want to measure the weight of the attribute "slope", then W(slope) is equal to 12/121 = 0.09. Here, 12 is the total number of times a slope occurred throughout the prediction procedure, and 121 is the sum of the weights of the eight attributes that were chosen and are presented in Table 3.

$$D(p) = \sum_{1}^{n} W(a_n) * R(a_n)$$

Here, 'D' represents the severity point of selected patient 'p' and 'R' represents the range of an individual attribute value. The range of an attribute is determined based on the severity level of significant features. For example, the attribute "thalach" range is divided into two categories. First, values between 60 and 100 are referred to as normal, while those beyond 100 are viewed severe. The $R(a_n)$ for an attribute is calculated using the categories mentioned above. The smart-contract function for predicting the severity of heart disease *via* feature computation approach is illustrated through Algorithm 6.

The algorithm for calculating the disease severity score assists in determining how serious the heart problem is. When a score exceeds a certain threshold, it is extremely serious. The patient information will now be seriously considered, and it will be given to the emergency care team.

### Emergency attribute based access control implementation

To supervise and monitor the diagnostic process during emergency situations, the organization initially constituted an Emergency Team (ET). ET members must be active participants of an organization. The healthcare members have prepared a list (L) of data

**Table 4 Defined rules for emergency access to EHR data.**

| Policy rule | Description |
|---|---|
| PR1 | The formation of ET This team must be an active member of an organization. |
| PR2 | Healthcare members must be approved members of an organization. |
| PR3 | Based on the preliminary investigation, ET should decide the emergency time duration. |
| PR4 | These medical personnel can only read patient information in an emergency. |
| PR5 | ET has rights to set the start time and End time of data access. |
| PR6 | The additional time has been allocated for the inclusion of treatment information by ET. |

**Table 5 Requested action satisfaction policy.**

| Operation permitted | Satisfaction policy |
|---|---|
| Read | PR2∧ PR3∧ PR4∧ PR5 |
| Insert | PR2∧ PR3∧ PR4∧ PR5∧ PR6 |

that must be accessible during an emergency (*Gardiyawasam Pussewalage & Oleshchuk, 2017*). Once the requested list has been created, the ET will review it to ensure that the request is accurate. Table 4 shows the number of policies that have been developed to prevent undesired and unregistered access to EHR data. Table 5 illustrates the policies developed for emergency situations to read and insert patient data.

Consider the following request: When there is a medical emergency, the medical team request access to the patient's record,

$$R_{req} = \{MID, T_1, T_2, Opr, PID\}$$

where, 'R' indicates a request for a particular record; the request needs to contain the following details so that it can obtain access to a record: Healthcare Requester Member Identity (MID), Start Time (T1), End Time (T2), Record Actions (Opr), and Patient Identity (PID). The ET must approve the timeframe's T1 and T2 based on the list (L) constructed during the preparation phase. The healthcare members (PR2) requesting a timeframe must be between T1 and T2. The request will be denied if it exceeds the time limit ($Time_{request} < T1 \&\& Time_{request} > T2$). There will be an additional time slot available for adding the new treatment information.

# SIMULATION SETUP AND EXPERIMENTAL RESULTS

The proposed system's development environment is composed of two parts: graphical user interface (GUI) part and network part. This simulation has been performed on Intel(R) Core (TM) i5-7200U CPU @ 2.50 GHz 2.71 GHz processor, 8 GB RAM, and 440 GB SSD HDD. On the Ubuntu 20.04 operating system, the network part has been created using Hyperledger fabric version 2.2.3 framework on Docker Engine. The implemented chaincode was written in NodeJS. AngularJS is used to develop the GUI component. The GUI interface offers many features such healthcare member login, patient registration,

record creation, accessing patient records by healthcare professionals, update records and view records. We deployed CouchDB to store world state information, LevelDB to store the transaction logs, and block data in the blockchain.

The system's performance has been evaluated using the Hyperledger Caliper framework (*Caliper, 2004*). The research has been done by looking at the number of users, transaction size, and transaction completion time. The performance evaluation is done against both invoke chaincode and ledger query functions. The proposed EHR blockchain system's performance estimation is compares to existing systems (*Chelladurai & Pandian, 2022*) to identify how network throughput, network latency, and resource consumption make a difference.

## Evaluation metrics

The metrics used to evaluate the efficiency of the proposed blockchain system and smart contracts are carried out in this section. The following essential parameters are considered in measuring the performance of the proposed system: success rate, average throughput, average latency and resource consumption. Five trials have been used to assess each metric. The following gives a quick explanation of each metric:

- Success rate: It is measured using the number of transactions executed successfully against the submitted transaction count.
- System throughput: The number of valid transactions that were committed to the blockchain system over a stipulated period is used to compute the success rate.
- System delay/latency: It is measured by how long it takes for the result to reflect in the blockchain system after a transaction has been submitted to the system.
- Resource utilization: It is determined from system's average CPU consumption percentage and memory used by the blockchain system.

## Security evaluations

The sharing of highly confidential communications puts patients and healthcare providers' privacy and security at severe risk, according to a review of the literature. Security and privacy fundamentals like confidentiality, integrity, and availability must be guaranteed by the system. Transaction layer security (TLS) is used by the Hyperledger Fabric to secure communication between network elements. The fabric system can function effectively without TLS, but it is not recommended in a production environment. In this system, TLS was implemented only in accordance with the system's security policy. Security, trust, and privacy needs are the ones that are the most unfulfilled. Privacy deals with the appropriate handling of sensitive data, particularly personal data. Security of the blockchain's records is mostly achieved through cryptography. Participants in a network also carry their private keys, which serve as personal digital signatures. Integrity and authentication are ensured by this digital signature. The TLS Protocol's main objective is to ensure data integrity and privacy. Showing how the proposed system ensures security is explained in further detail in the subsequent paragraph:

- **Confidentiality**—Confidentiality is possible by ensuring that the implementation uses a private blockchain and has constrained user access. Participants in this healthcare system will have various responsibilities and privileges. Reports kept in a private collection should be accessible only to the patient who authorized the diagnostic report and the concerned healthcare member. The authorized individual uses the certificate provided by the Fabric CA with a role attribute to identify themselves. Additionally, the TLS mechanism is enabled to guarantee the integrity and confidentiality of data sent between client application and network components.

- **Integrity & consistency**—Integrity refers to safeguarding private data against unauthorized modifications that may be intentional or accidental. This system makes use of blockchain technology, which ensures data integrity through an endorsement policy. Consistency is ensured with the ordering services. The transaction ordering is carried out by the ordering node of the Hyperledger Fabric network. Because the methods used in these systems, which depend on probabilistic consensus, ensure ledger consistency.

- **Availability**—A system's availability helps to ensure that it can be accessed and is fully functional whenever an authorized user needs to use it. Reduced failed connections to data in the blockchain are achieved by making the blockchain network fault tolerant. Multiple peers and three Orderer nodes are used in this system to achieve high availability.

Also, our access control system can withstand common network vulnerabilities, including the Sybil attack, Man-in-the-Middle attack, Race Condition attack, and Collusion attack.

- **Sybil attack:** This attack involves creating multiple fake identities to overwhelm a network. This system uses digital identities and authentication mechanisms to verify the identity of all participants. Additionally, the Fabric consortium blockchain uses consensus algorithms, the identities of the nodes are verified by the consensus protocol and the network is protected from such attacks.

- **Man-in-the-Middle attack:** This attack involves a malicious actor intercepting messages between two nodes. Hyperledger consortium blockchain verifies the identity of each node and encrypts the communication between them, making it impossible for a malicious actor to intercept messages. By verifying the identity of the user or entity, access is only granted to the user with the correct combination of attributes, thus enabling secure communication between parties.

- **Race Condition attack:** This attack involves two or more transactions being processed simultaneously to gain an advantage over other transactions. Hyperledger consortium blockchain uses Fabric CA and consensus protocols to ensure that transactions are processed in a secure and reliable manner. These protocols ensure that all nodes in the network process transactions in the same order and prevent any malicious node from taking control of the transaction process. Additionally, Hyperledger consortium
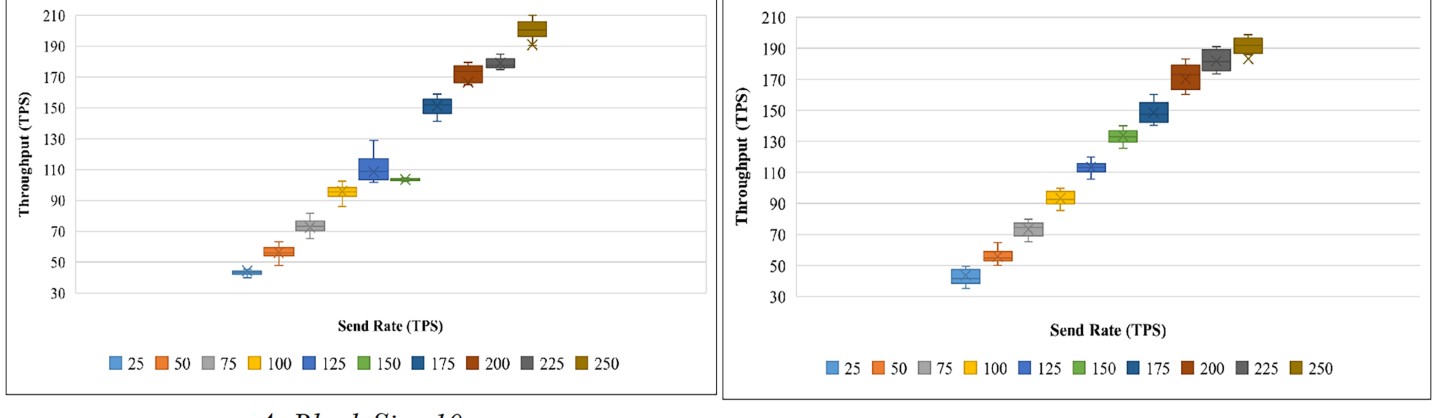

**A. Block Size 10.**

**B. Block Size 30.**

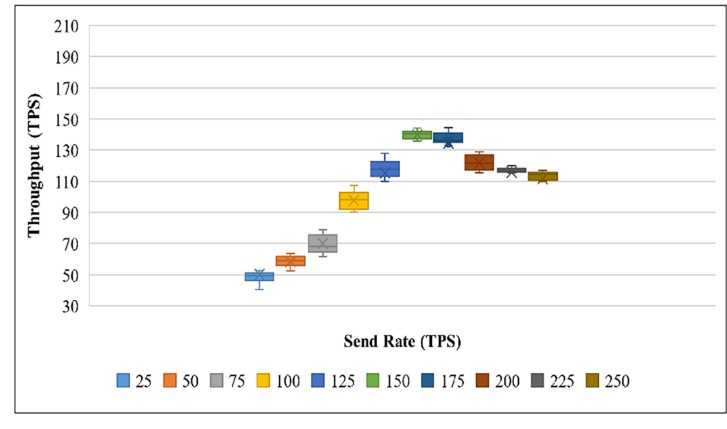

**C. Block Size 50.**

**Figure 5** **Evaluation of throughput.**

blockchain implements a smart contract feature which allows all participating nodes to validate the transactions and ensure that the data is secure.

- **Collusion attack:** Attribute based access control by ensuring that all attributes used in the access control decisions are unique. This can be done by assigning a unique identifier to each attribute. Additionally, all attributes should be cryptographically signed to ensure that they are not tampered with during the access control decision process. Finally, the system should be configured to reject any access control decision that results in a collision, *i.e.*, when two attributes of different owners have the same value.

## Performance assessment

To evaluate the performance of the proposed systems, the benchmark and network configuration files of Hyperledger Caliper were updated. With reference to block sizes 10, 30, and 50 and varied transaction send rates ranging from 25 to 250 tps, we analyzed the throughput and latency performance. We used a total of 10 different send rates to test the performance of the proposed system and conducted 20 attempts for each send rate. To

**Table 6 Evaluation of throughput.**

| Block size | TP (S) | Send rate (TPS) | | | | | | | | | |
|---|---|---|---|---|---|---|---|---|---|---|---|
| | | 25 | 50 | 75 | 100 | 125 | 150 | 175 | 200 | 225 | 250 |
| 10 | Min. | 38.6 | 47.9 | 52.7 | 67.9 | 101.8 | 102.6 | 141.5 | 165 | 175 | 190.7 |
| | Max. | 74.1 | 63.2 | 81.6 | 140.8 | 128.9 | 104.5 | 159.1 | 179.4 | 184.9 | 210 |
| | Median | 44.43 | 44.44 | 56.30 | 72.60 | 95.82 | 108.55 | 103.64 | 151.38 | 166.69 | 179.05 |
| 30 | Min. | 35.1 | 50.1 | 65.2 | 85.4 | 105.5 | 125.4 | 140.1 | 160.1 | 173.4 | 185.9 |
| | Max. | 49.5 | 64.7 | 79.9 | 99.8 | 119.9 | 139.9 | 160 | 183 | 191.1 | 198.7 |
| | Median | 43.62 | 55.87 | 73.45 | 93.12 | 113.37 | 133.54 | 148.41 | 170.19 | 181.76 | 183.13 |
| 50 | Min. | 40.5 | 52.6 | 61.6 | 90.3 | 110 | 135.6 | 133 | 115.5 | 115.3 | 110 |
| | Max. | 52.7 | 63.7 | 78.9 | 107.4 | 127.8 | 144.2 | 144.4 | 128.9 | 120 | 117 |
| | Median | 50.27 | 58.57 | 69.95 | 97.71 | 115.46 | 139.87 | 134.28 | 121.70 | 115.78 | 111.72 |

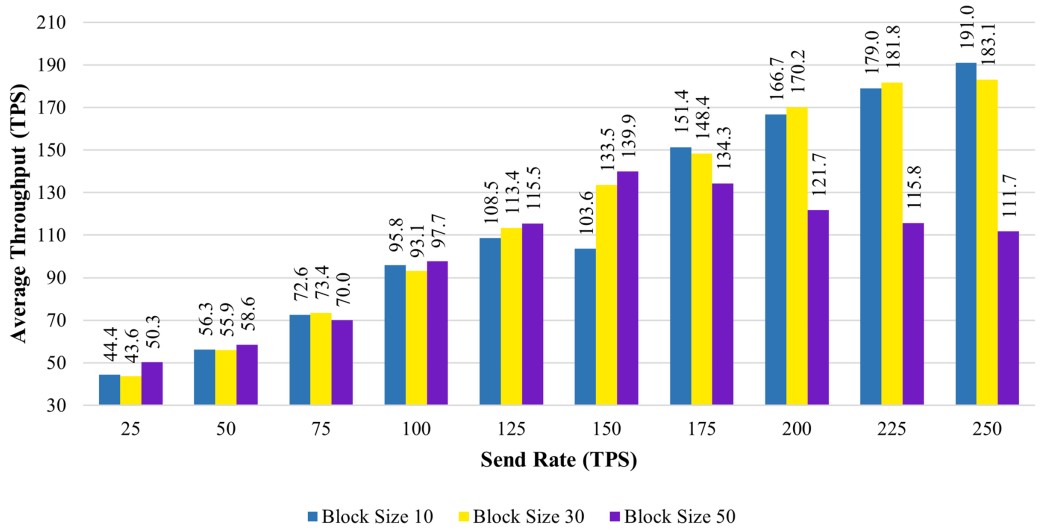

**Figure 6 Block size's impact on throughput evaluation.**

evaluate throughput, we have conducted a total of 200 trails (10 × 20). Here, 10 represents the overall send rate used, and 20 represents the number of trails for each send rate. When the experiment was being evaluated, we discovered a few outliers in the results. The box plot is used to determine the mean value after the outliers have been removed. A box plot is shown in Fig. 5A to show the range of throughput in relation to block size 10. Using the same procedure as before, we performed 200 trials with block sizes of 30 and 50. Figures 5B and 5C illustrate the throughput evaluation for blocks of 30 and 50, and Table 6 displays the summary of the obtained values.

Figure 6 shows the average transaction throughput statistics from the past investigation. The statistics revealed that for blocks having 10 and 30 transactions, the transaction throughput improved significantly with an increase in send rate. When the block size is

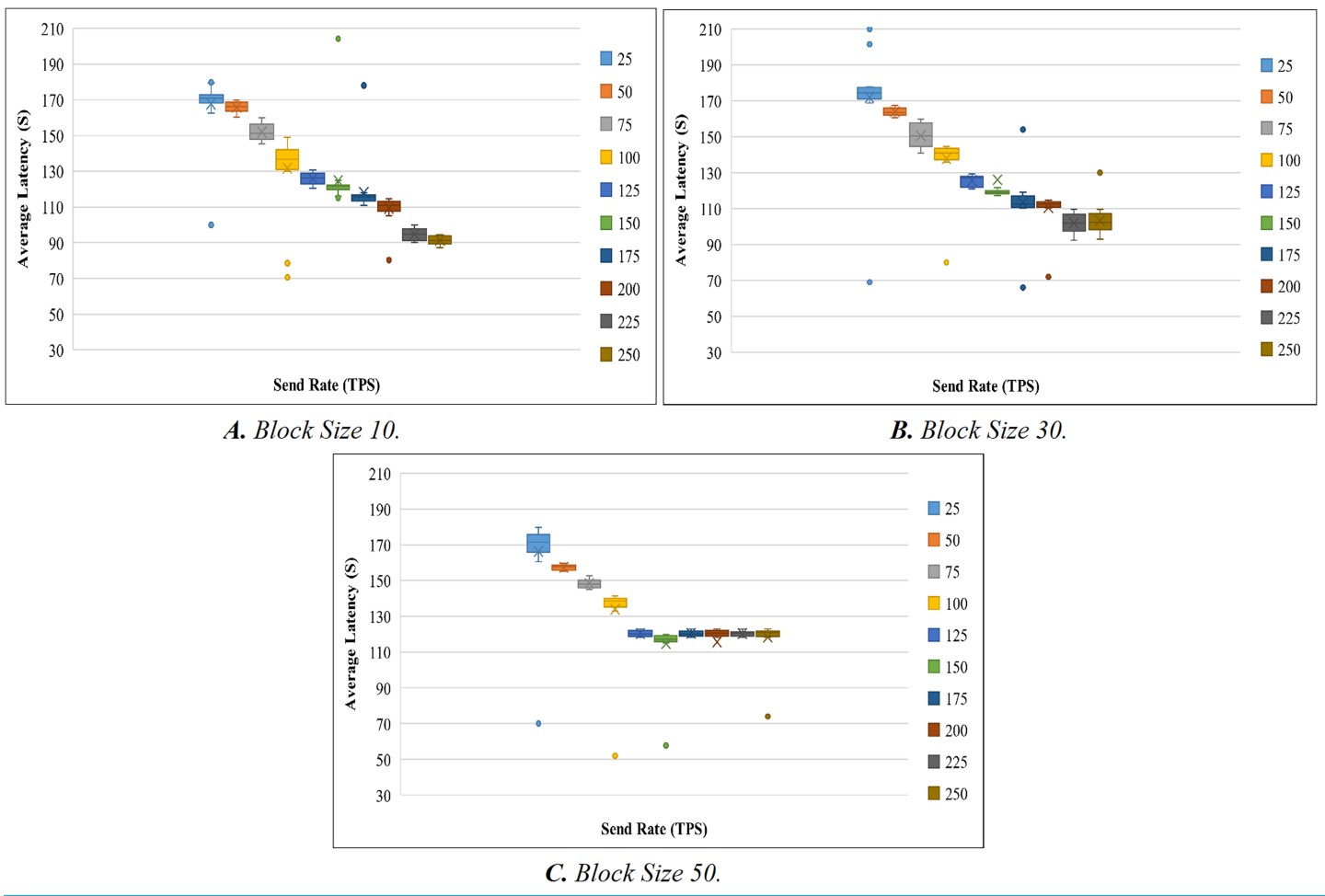

*A. Block Size 10.*

*B. Block Size 30.*

*C. Block Size 50.*

**Figure 7** Evaluation of latency.

increased to 50, the transaction throughput increases until the sending rate reaches 150 tps. As soon as the send rate exceeded 150 tps, the throughput rate declined.

Like the throughput evaluation, 200 trials have been completed for the latency investigation. The latency evaluation for blocks of 10, 30, and 50 is shown in Figs. 7A–7C, and Table 7 displays the summary of the obtained values.

Figure 8 shows the average transaction delay obtained in the experiment. Transaction latency decreases as the send rate rises from 25 tps to 250 tps for blocks with a size of 10 and 30. For blocks of size 50, the latency is initially reduced, and thereafter, consistency is preserved.

TLS's influence is considered while evaluating the proposed system's throughput and latency. Figures 9A–9C illustrates the comparison of throughput with and without TLS for blocks of 10, 30, and 50. When TLS was used, the transaction throughput improved exponentially as the send rate increased. On the other hand, the bar chart clearly indicates that throughput performance has dropped significantly in comparison to the non-TLS method. Figures 10A–10C illustrate a comparison of transaction latency with and without

**Table 7 Evaluation of latency.**

| Block size | TP (S) | Send rate (TPS) | | | | | | | | | |
|---|---|---|---|---|---|---|---|---|---|---|---|
| | | 25 | 50 | 75 | 100 | 125 | 150 | 175 | 200 | 225 | 250 |
| 10 | Min. | 100 | 160.2 | 145.5 | 70.6 | 120.4 | 115.1 | 110.8 | 80.2 | 90 | 87.2 |
| | Max. | 179.9 | 169.9 | 160 | 149.1 | 130.7 | 125 | 118 | 114.7 | 100 | 94.3 |
| | Median | 167.82 | 165.90 | 151.97 | 131.74 | 125.89 | 124.70 | 118.03 | 109.18 | 94.70 | 91.37 |
| 30 | Min. | 168.7 | 160.5 | 140.8 | 136 | 121 | 117.3 | 110.1 | 110.3 | 92.4 | 92.9 |
| | Max. | 210 | 167.5 | 159.8 | 144.7 | 129.4 | 121.6 | 119.1 | 114.7 | 109.6 | 130 |
| | Median | 171.89 | 163.93 | 150.47 | 137.77 | 125.48 | 126.06 | 113.25 | 110.50 | 101.87 | 103.31 |
| 50 | Min. | 160.6 | 155.1 | 145 | 133 | 118.1 | 115.2 | 118.1 | 118.1 | 118.1 | 118.1 |
| | Max. | 179.8 | 159.6 | 152.7 | 141.4 | 123 | 120 | 123 | 122.9 | 123 | 123 |
| | Median | 166.27 | 157.43 | 148.36 | 133.86 | 120.38 | 114.60 | 120.40 | 115.57 | 120.28 | 118.33 |

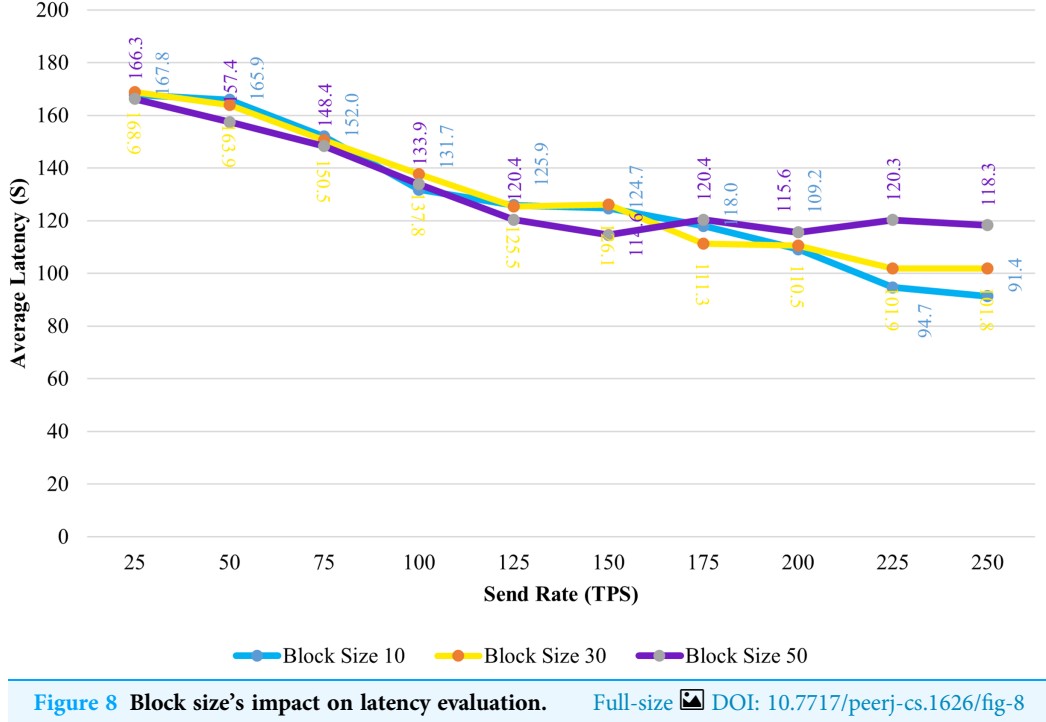

**Figure 8 Block size's impact on latency evaluation.**

TLS. Like transaction throughput, implementing TLS also had an impact on transaction latency performance. The outcomes of this experiment show that the use of TLS significantly affects performance. Even though it has a slight effect on system performance, TLS is required for private and sensitive data.

Based on the experimental analysis discussed above, we implemented this system in place with a block size of 30. In this system, a block can have a maximum of 30 transactions. Since the block size is 30 the system performs well in terms of throughput and latency in all the categories. Although there is a small difference in latency between networks with and without TLS. The results of the above experiment led us to the

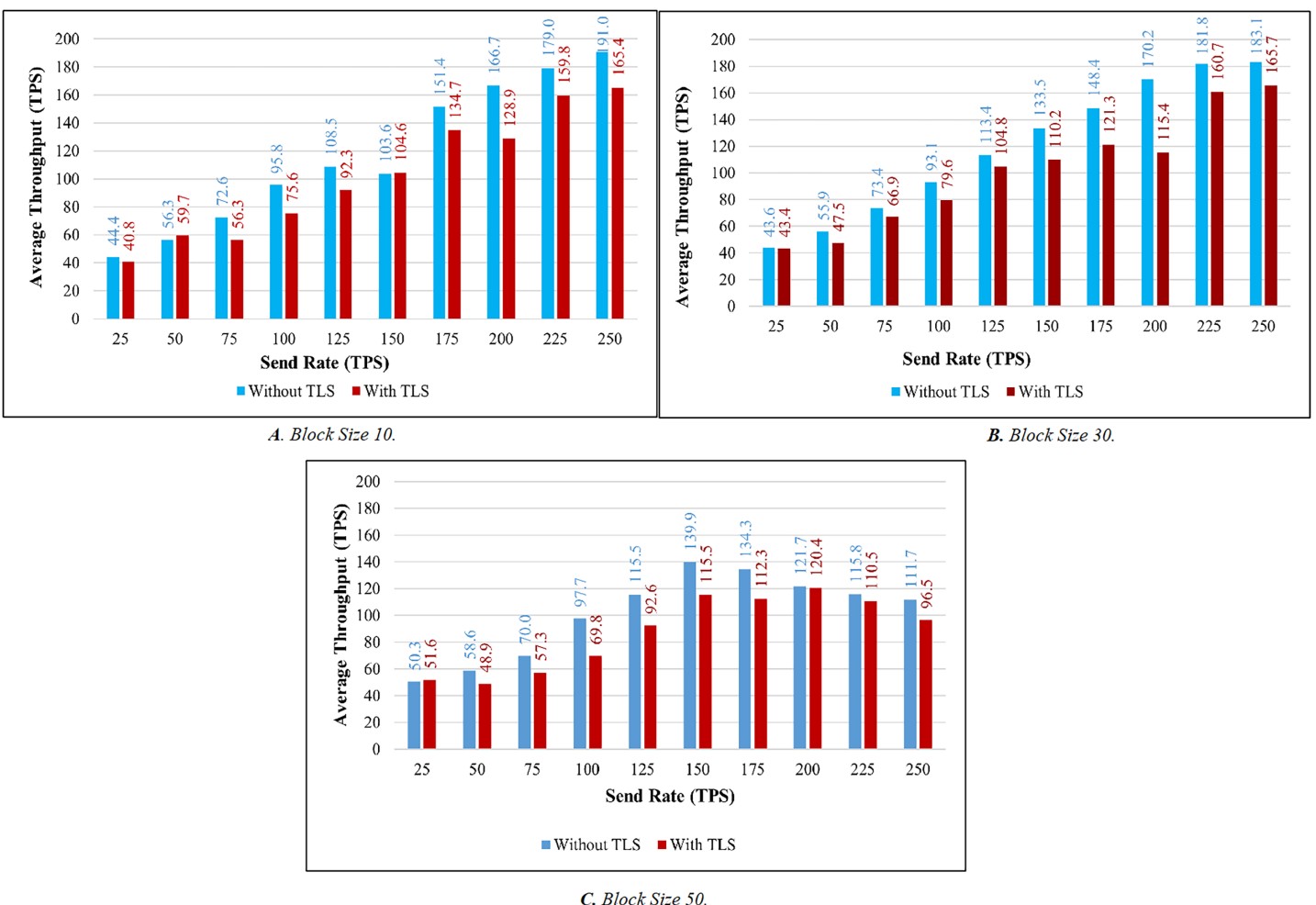

*A. Block Size 10.*

*B. Block Size 30.*

*C. Block Size 50.*

**Figure 9 Throughput evaluation of the impact of TLS.**

conclusion that TLS has little impact on system performance. As a result, we utilized TLS to ensure secure communication.

In Table 8, the proposed system's default configuration is provided. The specifics of this configuration are chosen in accordance with the results of earlier experiments. With the help of the above-mentioned configuration, multiple parallel transactions are submitted to determine the system performance in terms of throughput, latency, and resource consumption. There are various types of rate control mechanisms offered by the Hyperledger Caliper framework (*Caliper, 2004*). Here, we utilized two different rate control mechanisms to assess the efficacy of the proposed systems. First, the Fixed Load rate controller's aim is to maintain a defined number of transactions within the system. Second, the Linear Rate controller can gradually change the number of transactions in the system.

**Throughput:** Figure 11 demonstrates the throughput for 100 to 1,000 simultaneous transactions. The blue bar graph shows the throughput with a fixed load, and the brown bar graph shows the throughput using a Linear Rate. The system's throughput is

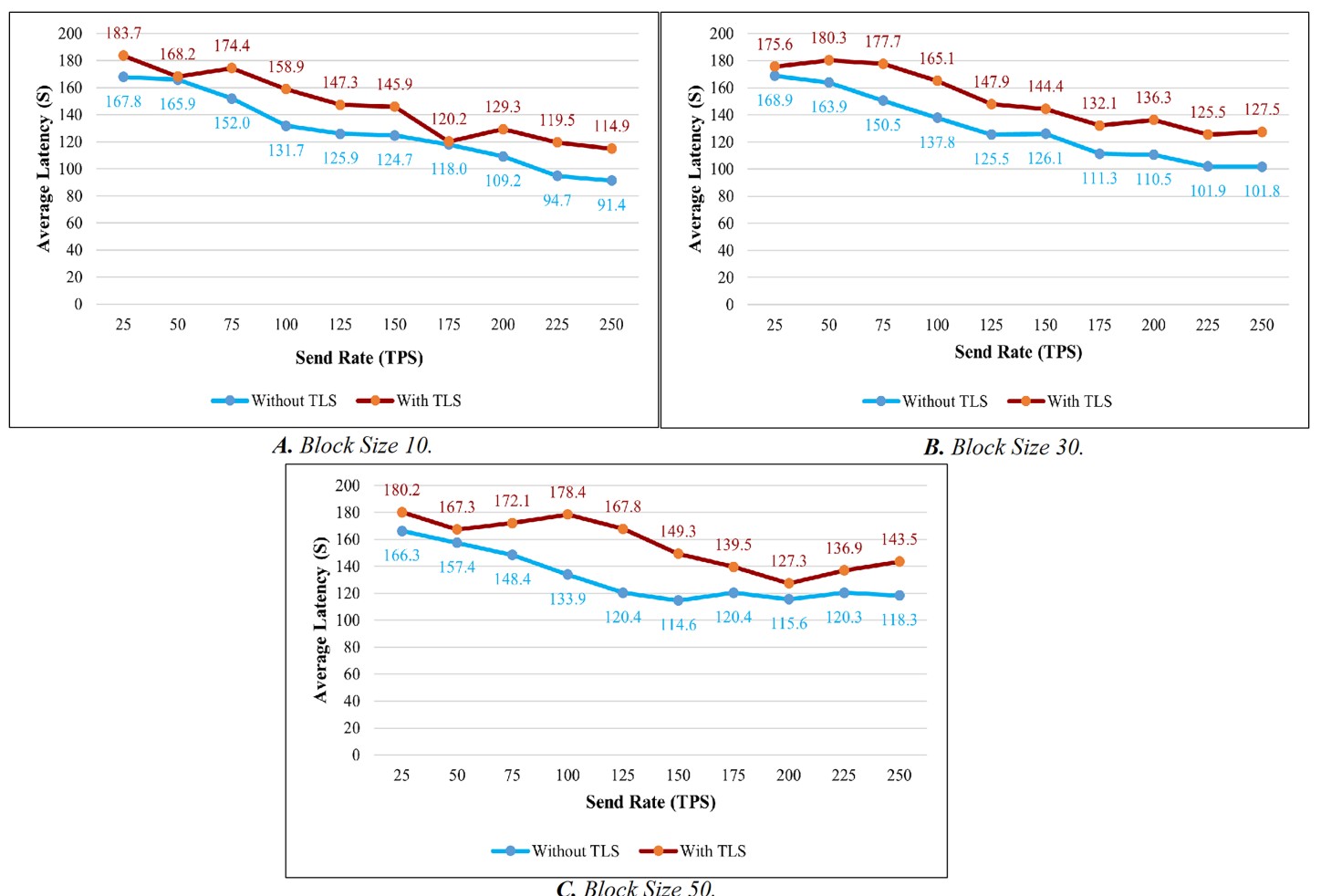

**A.** *Block Size 10.*

**B.** *Block Size 30.*

**C.** *Block Size 50.*

**Figure 10** Latency evaluation of the impact of TLS.

**Table 8 The default configuration for the system.**

| Network components | Values |
| --- | --- |
| Number of organizations | 2 |
| Number of endorser peers & committing peers | 2 |
| Ordering service | etcRaft |
| Block size | 30 transaction per block |
| TLS | Enabled |
| Endorsement policy | 1 of 4 (OR policy) |
| Programming language of smart contract | NodeJS |
| Send rate | 25–250 tps |

significantly higher than fixed load when it implements a linear rate control method. It has been shown that as the volume of transactions increases, both rate control strategies maximize the system throughput.

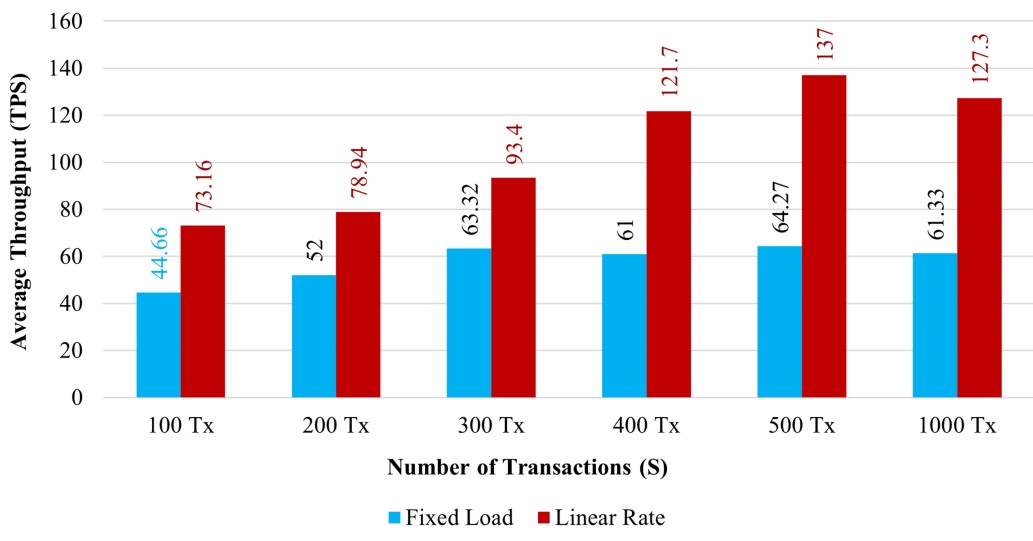

**Figure 11** **Transactions throughput on chaincode functions.**

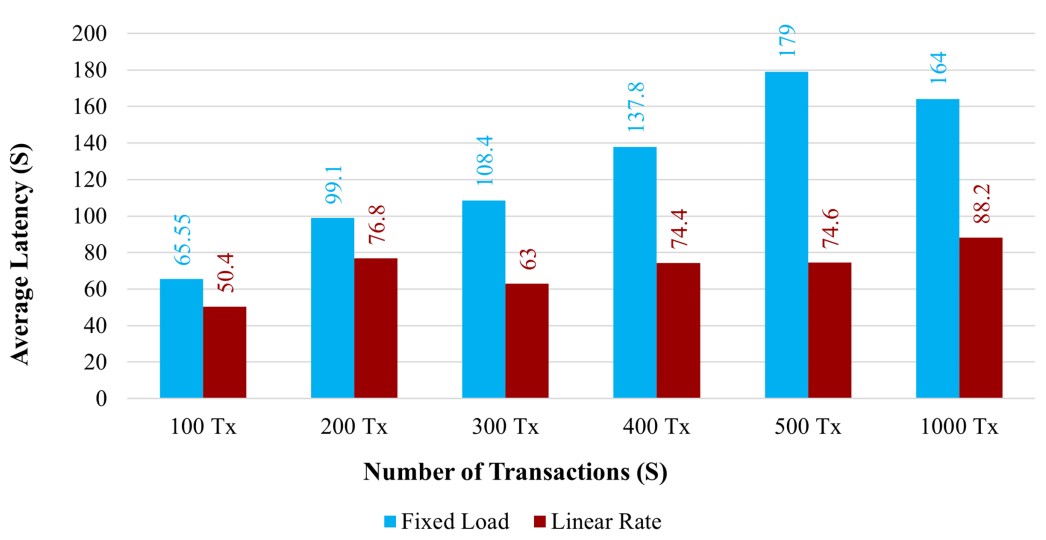

**Figure 12** **Average latency on chaincode functions.**

Throughput and response time of transactions are significantly associated. As the system increases overall throughput, the average transaction's response time tends to get faster. In comparison to the existing system, which requires a minimum of 2.5 s for a transaction to be completed. The proposed system completes over 81 transactions per second when we implement it using the linear approach. Then, one transaction can be completed by the present system in less than a second. According to the abovementioned observation, the proposed blockchain system has a higher throughput than the existing system.

**Latency:** Figure 12 demonstrates the average latency for 100 to 1,000 simultaneous transactions. The average latency using a fixed load is depicted in the blue bar graph, and the average latency using a linear rate is depicted in the brown bar graph. When the system

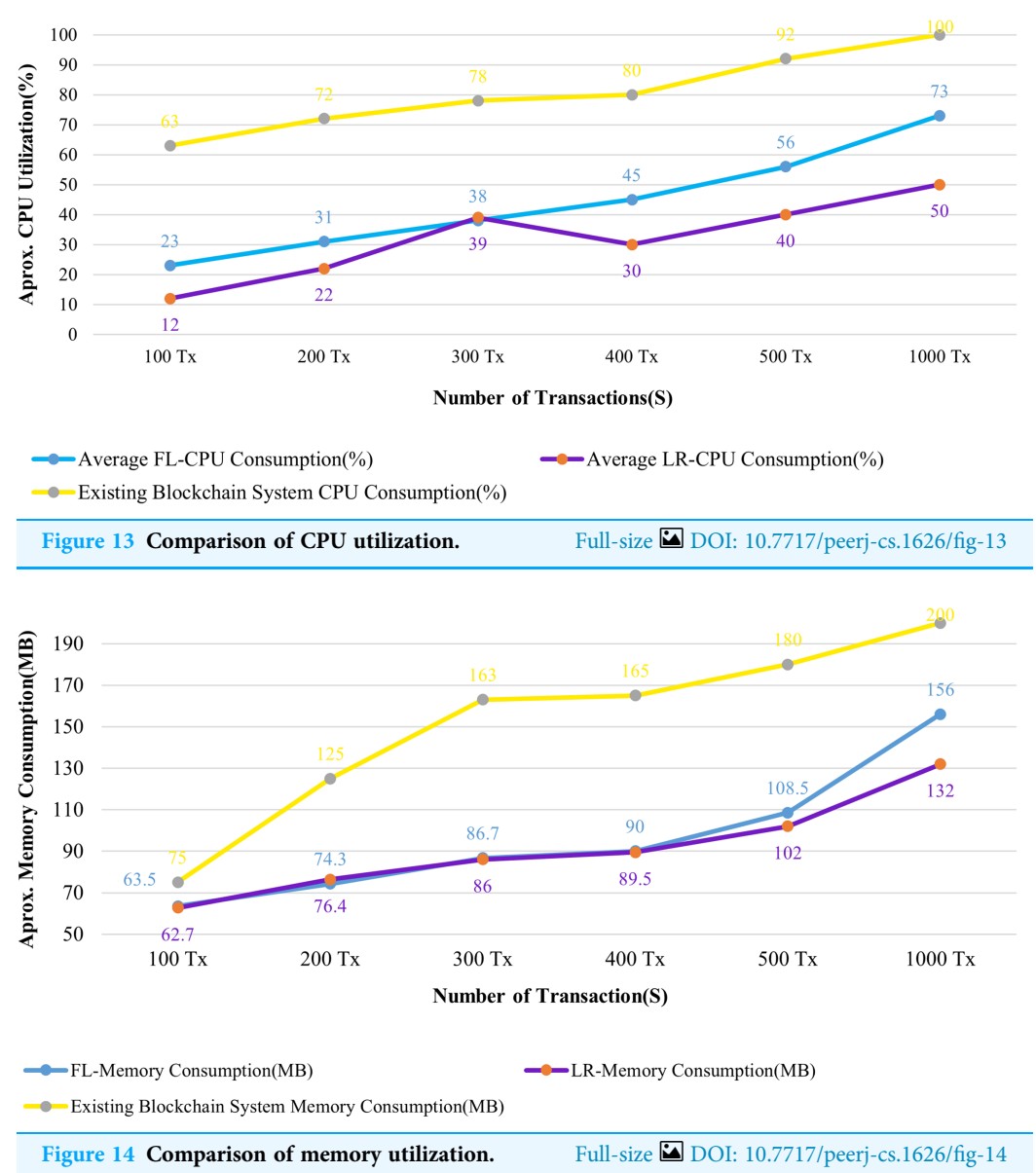

**Figure 13 Comparison of CPU utilization.** 

**Figure 14 Comparison of memory utilization.** 

uses a linear rate control strategy, its latency is increased compared with a fixed load approach. However, when the number of simultaneous transactions increased, the average latency also increased.

The overall amount of time needed for a transaction to take effect across the peers is known as the transaction delay. Comparing the proposed system to the existing one, the transaction delay is significantly increased. The average transaction time for 100 health records would take more time than it does under the existing system. The proposed system has a larger latency because there are more organizations and peers in each organization.

**Resource utilization:** The CPU and Memory usages of the system are analyzed in connection with 100 to 1,000 patient records that have been submitted to the blockchain system.

Figures 13 and 14 evaluate the proposed system to an existing blockchain system in terms of CPU and memory consumption. We evaluated two distinct rate controller mechanisms offered by Hyperledger Caliper in this instance to assess how well the suggested system performed. From the above evaluation, this system requires, on average, 63 MB of storage for 100 transactions of health records, compared to 75 MB for the existing system. In terms of CPU utilization, this system uses a maximum of 18% for 100 health records transactions, whereas the existing system uses 63% for 100 health records transactions. It is clear from both instances that the proposed system uses less CPU and memory than the existing blockchain system for all categories of transactions. As a result, the fabric based EHR blockchain system uses very less CPU than the existing blockchain system while maintaining storage consistency between the systems. Along with the benefits, TLS also contributes to a significant improvement in privacy and security.

## CONCLUSION

Blockchain is conscious to extend the trust to share the data among multiple participants of the network. Using Blockchain technology, this research has proposed a solution for achieving privacy and transparency of Electronic Health Records. This system facilitates attribute-based access control mechanisms and private data conceptualization to restrict EHR access by illegal users. As a result, patients have entire authority over their medical records, and they may choose who can access them. Accordingly, the experimental network has been designed and configured for privacy and security. In addition to the above, the security measures added by the TLS protocol include authentication and message tampering detection between the client and participating nodes. This proposed system integrates a Gaussian naïve Bayes classifier for heart disease prediction. This algorithm has been used to identify heart disease since it can reliably determine whether a patient has or does not have heart disease based on the values of the features in the patient's data set. Thereafter, the feature weight computation technique was utilized to determine the severity level by assigning different weights to each feature to express its strength. Emergency attribute-based access control allows authorized personnel to access the patient data in a timely manner. This helps ensure that the patient is provided with the best possible care in emergency situations. Finally, the performance of the proposed system has been evaluated in terms of success rate, latency, throughput and resource utilization. Hyperledger Caliper's two different rate controller mechanisms were used to evaluate this system. The result shows that, in terms of throughput and resource utilization, performance has increased in comparison to the existing system.

In the future, we plan to integrate medical image sharing into the system, allowing medical professionals to securely share images such as X-rays, CT scans, MRI scans, ultrasounds, and other medical images. Storing large amounts of data on a blockchain is not only very expensive, but it can also cause the blockchain to become slower and less reliable. To solve those issues optimization techniques can be used to improve the system performance.

### Funding

The authors received no funding for this work.

### Competing Interests

The authors declare that they have no competing interests.

### Author Contributions

- Sasikumar R. conceived and designed the experiments, performed the experiments, analyzed the data, performed the computation work, prepared figures and/or tables, authored or reviewed drafts of the article, and approved the final draft.
- Karthikeyan P. conceived and designed the experiments, analyzed the data, authored or reviewed drafts of the article, and approved the final draft.

### Data Availability

The Heart Disease Dataset is available at Kaggle: https://www.kaggle.com/datasets/johnsmith88/heart-disease-dataset.

The Diabetes Dataset is available at Kaggle: https://www.kaggle.com/datasets/mathchi/diabetes-data-set.

The Stroke Prediction Dataset is available at Kaggle: https://www.kaggle.com/datasets/fedesoriano/stroke-prediction-dataset.

The Body Fat Prediction Dataset is available at Kaggle: https://www.kaggle.com/datasets/fedesoriano/body-fat-prediction-dataset.

### Supplemental Information

Supplemental information for this article can be found online at http://dx.doi.org/10.7717/peerj-cs.1626#supplemental-information.

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
