# Peer review of "Heart disease severity level identification system on Hyperledger consortium network"

_PeerJ Computer Science, doi:10.7717/peerj-cs.1626_

## Round 0.1 · original submission · Major Revisions

Dear Authors,
Good Work. Please complete these suggestions to improve the quality of the article, then resubmit for the final review.

Reviewer 1 ·

Basic reporting

The manuscript is well written and has sufficient potential to be published in the journal.

Experimental design

The work is novel. They provide the steps of the registration clearly.

There are some typos to be corrected:
Line 80 - "responsible" is written instead of "responsibility"
Line 203 - HER instead of EHR
Line 297 - Use CA_Public Key as sub_index of Encr function, as used in other steps

Also 2 more questions:
Line 279 - What kind of key is RK? Is it symmetric or asymmetric? It is used later on line 306 to make encryption.
Line 287 - Why PK refers the secret key? Is it a typo?

It might be better to use standard citation method with numbers in brackets ([1], [1,2], [1,4], etc.) instead of using the author names directly in the sentence unless they are not the subject of the sentece. Your way harms the integrity of the sentence. Or you can try using it in brackets.

Validity of the findings

The authors are offering a block-chain based EHRs sharing platform with the following components:
- Hyperledger Consortium Network for sharing health records
- ABAC is used for controlling access to EHRs
- Gaussian Naive Bayes Algorithm for prediciton of cardiovascular disease

The authors have provided sufficient evidence to show the results of their research.

Reviewer 2 ·

Basic reporting

In this paper, author presented a Hyperledger consortium network has been developed for sharing health records with enhanced privacy and security. Via the designed experiments, the efficiency and effectiveness of the proposed method has been proven. The discussed issue shows the research significance, and the structure is relative good. However, major revisions are needed before the acceptance.


1. How to deploy the local database systems to establish the secure connection with blockchain sytems..
2. What consensus algorithms are used in the integrated blockchain system to provide the acceptable performance.
3. To maintain the blockchain system running in this work, how to deploy the full nodes and lightweight nodes.
4. Please go through the paper carefully and double check whether the right template are used. Correct some typos and formatting issues (e.g., “User Identity and Enrolment process” -> “User Identity and Enrolment Process”?).
5. The writing should be well improved, and the Figure 3 and 4 can be presented in other ways.
6. A notation table can be provided to improve the readability.
7. Make the References more comprehensive, besides this work, some other promising scenarios (e.g., Big data, other IoT systems) can be covered in this work. If the above related work can be discussed, it can strongly improve the research significance.

Experimental design

see report

Validity of the findings

see report

Additional comments

see report

Reviewer 3 ·

Basic reporting

Make the list of references more complete. This work can also be used to talk about some other potential ideas. If the above linked work can be talked about, it can make the study much more important. For the improvement, the following works can be looked at to make the sources more complete.

Experimental design

How to install full nodes and lightweight nodes to keep the blockchain system working in this work.
How to set up a secure connection with blockchain systems using local database systems.
What agreement methods does the combined blockchain system use to make sure it works well?

Validity of the findings

Please read the material carefully and double-check that the correct template is used. Correct a few typos and formatting errors.
The text should be enhanced, and Figures 3 and 4 may be presented in other ways.

Additional comments

For better readability, a notation table may be given.

---

## Round 0.2 · accepted · Accept

Dear Author/Authors,

Greetings for the Day!!!

Thank you for completing the requested modification as per the peer review process.

Once again congratulations for the good work contributed to society and proceed to the next steps.

Thanks
Dr. M. Nageswara Rao
AE-PeerJ Computer Science

Reviewer 1 ·

Basic reporting

No comment

Experimental design

No comment

Validity of the findings

No comment

Additional comments

I think the paper is revised based on previous review and the revision should be accepted.

Reviewer 2 ·

Basic reporting

The authors have addressed my previous concerns. It can be accepted now.

Experimental design

The authors have addressed my previous concerns. It can be accepted now.

Validity of the findings

The authors have addressed my previous concerns. It can be accepted now.

Additional comments

The authors have addressed my previous concerns. It can be accepted now.

Reviewer 3 ·

Basic reporting

overall paper is good
validation is found, even reproducibility of results can be promised

Experimental design

good

Validity of the findings

validation is found, even reproducibility of results can be promised
I do find this paper suitable enough to publish in PeerJ Computer Science journal